# Pharmacological Modulation of the Unfolded Protein Response as a Therapeutic Approach in Cutaneous T-Cell Lymphoma

**DOI:** 10.3390/biom15010076

**Published:** 2025-01-07

**Authors:** Nadia St. Thomas, Benjamin N. Christopher, Leticia Reyes, Reeder M. Robinson, Lena Golick, Xiaoyi Zhu, Eli Chapman, Nathan G. Dolloff

**Affiliations:** 1Department of Pharmacology and Immunology, Medical University of South Carolina, 173 Ashley Ave., MSC509, Charleston, SC 29425, USA; stthoman@musc.edu (N.S.T.); christob@musc.edu (B.N.C.); reyesl@musc.edu (L.R.); robinree@musc.edu (R.M.R.); golick@musc.edu (L.G.); 2Department of Pharmacology and Therapeutics, Center for Inflammation Science and Systems Medicine, University of Florida Scripps Institute for Biomedical Innovation and Technology, Jupiter, FL 33458, USA; xiaoyizhu@ufl.edu (X.Z.); chapmaneli@ufl.edu (E.C.); 3Hollings Cancer Center, Medical University of South Carolina, Charleston, SC 29425, USA; 4Zucker Institute for Innovation Commercialization, Charleston, SC 29425, USA

**Keywords:** cutaneous T-cell lymphoma, ER stress, unfolded protein response, bortezomib, HSP70, HSPA5, HSPA6

## Abstract

Cutaneous T-cell lymphoma (CTCL) is a rare T-cell malignancy characterized by inflamed and painful rash-like skin lesions that may affect large portions of the body’s surface. Patients experience recurrent infections due to a compromised skin barrier and generalized immunodeficiency resulting from a dominant Th2 immune phenotype of CTCL cells. Given the role of the unfolded protein response (UPR) in normal and malignant T-cell development, we investigated the impact of UPR-inducing drugs on the viability, transcriptional networks, and Th2 phenotype of CTCL. We found that CTCL cells were >5-fold more sensitive to the proteasome inhibitor bortezomib (Btz) and exhibited a distinct signaling and transcriptional response compared to normal CD4+ cells. The CTCL response was dominated by the induction of the HSP70 family member *HSPA6* (HSP70B’) and, to a lesser extent, *HSPA5* (BiP/GRP78). To understand the significance of these two factors, we used a novel isoform selective small-molecule inhibitor of HSPA5/6 (JG-023). JG-023 induced pro-apoptotic UPR signaling and enhanced the cytotoxic effects of proteasome inhibitors and other UPR-inducing drugs in CTCL but not normal T cells. Interestingly, JG-023 also selectively suppressed the production of Th2 cytokines in CTCL and normal CD4+ T cells. Conditioned media (CM) from CTCL were immunosuppressive to normal T cells through an IL-10-dependent mechanism. This immunosuppression could be reversed by JG-023, other HSP70 inhibitors, Btz, and combinations of these UPR-targeted drugs. Our study points to the importance of the UPR in the pathology of CTCL and demonstrates the potential of proteasome and targeted HSPA5/6 inhibitors for therapy.

## 1. Introduction

Cutaneous T-cell lymphoma (CTCL) is characterized by the infiltration and expansion of malignant clonal T lymphocytes in the skin. Skin patches and plaques are pruritic and painful and may progress to confluent erythema that can ulcerate and affect >80% of the body surface area [1]. CTCL is a heterogenous group of malignancies, with Mycoses fungoides (MF), Sézary Syndrome (SS), and primary cutaneous CD30+ anaplastic large cell lymphoma (pcALCL) representing 80–85% of cases [2]. MF remains localized to the skin, whereas the more aggressive leukemic form, SS, is found in the peripheral blood and lymph nodes and carries a significantly worse prognosis. Early-stage MF is a treatable and indolent disease, whereas advanced MF and SS are incurable, debilitating, and more aggressive, with a median survival estimated at less than 1.5 years and poor health-related quality of life [3,4]. For these patients, topical therapies are ineffective, and the standard of care includes the retinoid analog, bexarotene, interferon-α, HDAC inhibitors (vorinostat/ZOLINZA and romidepsin/ISTODAX), and, more recently, the development of monoclonal antibodies, brentuximab/ADCETRIS and mogamulizumab/POTELIGEO, which target CD30 and CCR4, respectively. These newer agents have shown promising response rates. However, they are short-lived, and a large international study showed that survival rates have not increased in the U.S. despite the emergence of these new drugs [5]. Therefore, there is a need for innovative new treatments for CTCL, which necessitates the identification of novel therapeutic targets through a more complete understanding of the fundamental cellular and molecular disease underpinnings.

Genetic analysis has identified mutational events that are candidate drivers of MF and SS, including loss of tumor suppressors and epigenetic regulators and dysregulation of the NF-κB, MAPK, and JAK signaling pathways [6,7,8]. These studies shed light on the intrinsic events that influence cell-autonomous cell growth, survival, and proliferation. However, much less is known about the extrinsic factors in the tumor microenvironment (TME) that spur disease onset, progression, and therapeutic resistance. The CTCL TME, like that of other tumor types, is complex and unique to the organ site of the primary tumor and the repertoire of cell types that mediate bilateral communication between malignant cells and normal bystanders. The architecture of the CTCL TME is dictated by the anatomy of the dermis which supports skin-resident keratinocytes, Langerhans cells, melanocytes, and fibroblasts, along with immune cell populations, including non-malignant T and B cells, dendritic cells, neutrophils, NK cells, and others [9,10,11,12]. The immunophenotype of MF and SS is predominantly CD4+, with MF showing characteristics of skin-resident CD4+ effector memory T cells and SS more closely resembling circulating central memory CD4+ T cells [13]. Malignant T cells from CTCL lesions show a strong type 2 T helper (Th2) bias characterized by the production of Th2 cytokines (Il-4, IL-5, IL-10, IL-13) and the low production of Th1 cytokines (IFNγ, TNFα, and IL-2) [9,14]. Immunologic abnormalities observed in patients that are characteristic of a Th2-dominant imbalance include T-cell and NK-cell suppression, increased eosinophilia, and increased serum IgE and IgA [15,16], M2 macrophage polarization, suppression of proinflammatory cytokine secretion, and T-cell anergy [17,18,19]. Patients with advanced disease exhibit a generalized state of systemic immunosuppression and are highly susceptible to cutaneous infections, bacteremia, and pneumonia which are leading causes of death [20,21,22,23]. Little is known about the disease-specific regulators that govern the Th2 phenotype of CTCL or how this affects disease progression and clinical outcomes. Investigation in this area could offer a more complete understanding of the disease and lead to new therapeutic strategies that restore immune balance and resolve CTCL lesions.

The unfolded protein response pathway (UPR) is a multifaceted cellular response induced by an increased cellular demand for protein production, folding, and modification [24]. When challenged by stress signals from misfolded proteins in the endoplasmic reticulum (ER), UPR signaling initially protects the cell from proteotoxic damage and then induces cellular changes that allow for increased protein load. Adaptations include pausing of further protein translation in the short-term through the inhibition of cap-dependent translation, followed by genetic changes that impact protein folding, degradation, glycosylation, ER-Golgi transport, and the induction of protein chaperone and heat shock protein (HSP) transcription [25]. On the other hand, UPR also signals apoptotic cell death with extensive and sustained ER stress. Studies have shown that UPR signaling is also an important component of the differentiation program of B and T cells. For example, the differentiation and maturation of B lymphocytes into antibody-secreting plasma cells is dependent upon the X-box Binding Protein-1 (XBP-1) arm of the UPR [26,27]. Likewise, evidence suggests that XBP-1 and its upstream activator, inositol-requiring enzyme 1α (IRE1α), are important for helper T-cell function and the differentiation of Th2 CD4+ helper T cells specifically [28,29]. Altogether, these reports support a role for UPR effector signaling in normal T-cell development, although, exactly how UPR signaling impacts the phenotypic diversity is not well understood. Furthermore, whether signaling is maintain or divergent in malignant CTCL T cells remains unknown.

In this study, we discovered that normal CD4+ T cells and CTCL cells exhibit distinct UPR signaling in response to pharmacological ER stressors. Using the proteasome inhibitor, bortezomib (Btz), as a model compound, we show that CTCL cells rely predominantly on the PERK-dependent induction of specific Hsp70 family members, HSPA5 (BiP; GRP78) and HSPA6, for protection against toxic levels of ER stress. We further demonstrate, using a selective inhibitor of HSPA5/6 (JG-023), that the production of Th2 cytokines, but not Th1 cytokines, is HSPA5/6-dependent in both normal and malignant T cells. Conditioned media from CTCL cells directly and indirectly suppress normal T-cell activation, and this immunosuppressive activity can be reversed using novel HSPA5/6-selective inhibitors. Our work offers new insight into the control of the Th2-dominant immunosuppressive effects of CTCL and suggests that selective HSPA5/6 inhibitors could be effective targeted therapies for the treatment of CTCL.

## 2. Materials and Methods

### 2.1. Cell Lines and Reagents

HH and Hut78 CTCL cell lines were purchased from the American Tissue Culture Collection (Manassas, VA, USA). Primary human peripheral blood mononuclear cells (PBMCs) were purchased from STEMCELL Technologies (Cambridge, MA, USA). Bortezomib (Catalog No. S1013), VER155008 (Catalog No. 57751), CB5083 (Catalog No. S8101), pevonedistat/MLN4924 (Catalog No. S7109), ricolinostat (Catalog No. S8001), alvespimycin/17-DMAG (Catalog No. S1142), GSK2606414 (Catalog No. S7307), and phorbol 12-myristate 13-acetate (Catalog No. S7791) were purchased from Selleck Chemicals (Houston, TX, USA). Ionomycin (Catalog No. 73722) was purchased from STEMCELL technologies (Cambridge, MA, USA). JG-023 was provided by Dr. Eli Chapman (U. of Arizona) and has been previously described [30].

### 2.2. Cell Culture

HH and Hut78 were serially passaged in RPMI 1640 with 2.05 mM L-Glutamine (Cytiva; Marlborough, MA, USA Catalog No. SH30027.01) supplemented with 10% heat-inactivated fetal bovine serum (FBS) (GeminiBio; Liverpool, UK Catalog No. 900-108) and antibiotics [1% penicillin (10,000 units/mL), 1% streptomycin (10,000 µg/mL), and 1% amphotericin B (25 µg/mL)] (Cytiva; Marlborough, MA, USA Catalog No. SV30079.01). PBMCs were stimulated with Immunocult Human CD3/CD28/CD2 T Cell Activator (STEMCELL technologies, Catalog No. 10970), and CD4+ T cells were isolated using the EasySep Human CD4+ T cell Enrichment Kit (STEMCELL technologies, Catalog No. 19052) according to the “The Big Easy” EasySep Magnet protocol (STEMCELL technologies, Catalog No. 18001). PBMCs and isolated CD4+ T cells were cultured in Immunocult- XF T cell Expansion Media (STEMCELL technologies, Catalog No. 10981) and serially passaged with the addition of CD3/CD28/CD2 T cell activator, 10 µL/1 × 10^6^ cells and human IL2, 0.6 µL/1 × 10^6^ cells (TECIN Teceleukin, National Cancer Institute; Frederick, MD, USA Catalog No. 23-6019).

### 2.3. Flow Cytometry

Stimulated PBMCs and CD4+ T-cell isolates were collected, washed with 1 mL of sterile flow buffer (1X PBS pH 7.4 with 150 µM of CaCl_2_ and 1% FBS), and centrifuged at 2500 rpm 4 °C for 5 min. Cells were resuspended in 100 µL of flow antibodies prepared at a 1:50 dilution with FACs buffer and incubated on ice for 30 min protected from light. FITC mouse IgG1 kappa Isotype Control (Catalog No. 555748), FITC mouse anti-human CD4 (Catalog No. 555346), and FITC mouse anti-CD8 (Catalog No. 555634) were purchased from BD Pharmigen (San Jose, CA, USA). Live/dead staining was run using propidium iodide (Invitrogen, ThermoFisher Scientific; Waltham, MA, USA Catalog No. P3566). Following incubation, cells were washed with 1 mL of FACs buffer and resuspended in 300 μL of FACs buffer for analysis. For cleaved (active) caspase-3 staining, cells were washed with ice-cold PBS and fixed with the Fixation/Permeabilization Solution Kit from BD Biosciences (San Jose, CA, USA) according to the manufacturer’s instructions. Cells were then treated with a 0.125 mg/mL final concentration of rabbit anti-active caspase-3 (BD Pharmigen; San Jose, CA, USA) and incubated at room temperature for 20 min. Cells were then washed with permeabilization/wash solution and resuspended in 50 µL of diluted Alexa Fluor 488 goat anti-rabbit IgG (Invitrogen; Carlsbad, CA, USA) according to the manufacturer’s instructions. The cells were incubated for 20 min at room temperature while protected from light, washed with permeabilization/wash solution, and resuspended in 300 µL of permeabilization/wash solution and analyzed. All samples were analyzed with the NovoCyte flow cytometer (ACEA Biosciences; San Diego, CA, USA) and data were analyzed using FlowJo version 10.10.0 software.

### 2.4. Western Blot

HH, Hut78, and CD4+ T cells treated as indicated were collected on ice, rinsed with cold 1X PBS, centrifuged at 2500 rpm at 4 °C for 5 min, and lysed in 1X cell lysis buffer (Cell Signaling Technologies, Danvers, MA, USA; Catalog No. 9803) with the addition of protease (Catalog No. A32955) and phosphatase (Catalog No. A32957) inhibitors purchased from ThermoFisher Scientific (Waltham, MA, USA). Cell lysates were clarified and their relative protein concentrations were determined via a Bradford assay using Bio-Rad Protein Assay Dye Reagent Concentrate diluted 1:5 with MiliQ water (Bio-Rad Laboratories; Hercules, CA, USA Catalog No. 5000006). Values were normalized to the lowest average absorbance at 595 nm. Gel samples were prepared by mixing lysates with SDS sample buffer containing β-mercaptoethanol (final concentration 1X) and 1X cell lysis buffer. Samples were boiled for 10 min, loaded on NuPAGE Bis-Tris Gel 4–12% (Invitrogen Catalog No. NP0336BOX), and electrophoresed at 55 mA for 1 h and 45 min in 1X NuPAGE MOPS SDS Running Buffer (Invitrogen, ThermoFisher Scientific; Waltham, MA, USA Catalog No. NP0001). Gels were transferred to polyvinylidene difluoride (PDVF) membranes at 300 mA for 2 h in 1X Transfer Buffer containing 25 mM trizma base, 192 mM glycine, and 20% methanol. PDVF membranes were blocked for 1 h at room temperature with 5% MILK in TBS Tween prior to incubation with primary antibodies in 5% MILK TBS Tween overnight at 4 °C. Primary and secondary antibodies are listed below in Table 1. Detection was finalized using ECL (Catalog No. 32209) or Super Signal (Catalog No. 34094) detection reagents from ThermoFisher Scientific (Waltham, MA, USA).

### 2.5. Rt-qPCR

Total RNA was extracted from treated cells for quantitative reverse transcriptase polymerase chain reaction (RT-qPCR) using the Qiagen RNeasy Plus Mini Kit (Catalog No. 74134) per the manufacturer’s instructions. Isolated RNA concentrations were determined via NanoDrop, and RNA was reverse transcribed using the Luna Universal One-Step RT-qPCR Kit (New England BioLabs; Ipswich, MA, USA Catalog No. E3005L). Samples were run using the QuantStudio 3 Real-Time PCR System (AppliedBiosystems, ThermoFisher) and data were analyzed with QuantStudio3 qPCR Data Analysis. Triplicate raw C_t_ values for the housekeeping gene (GAPDH) and genes of interest were averaged and used to calculate the ΔΔC_t_ values. Changes in gene expression were evaluated as a measure of fold change (2^−ΔΔCt^). The primer sequences used for each of the gene targets are shown below in Table 2.

### 2.6. Cell Viability Assays

CellTiter-Glo^®^ (CTG) (Promega; Madison, WI, USA Catalog No. G9683) and a Spectramax L microplate luminometer (Molecular Devices; San Jose, CA, USA) were used to measure the cell viability of HH, Hut78, and CD4s treated as indicated. Following the 48 h incubation at 37 °C and 5% CO_2_, CTG was added to each well, plates were incubated at 37 °C and 5% CO_2_ for 5 min, and then, plates were read at 470 nm. Data were analyzed in Excel and the cytotoxicity of the compounds was expressed as the percentage cell viability compared to the control. The percent cell viability data were analyzed in GraphPad PRISM version 10.4.1. For experiments evaluating drug synergy, viability data were normalized to values in the absence of the drug 2 (Btz, carfilzomib) to account for cell death induced by single-agent drug 1 (JG-023, VER155008, AP-4-139B), as described previously [31,32]. By using this normalization method, any separation of the drug 2 dose response curves indicates a true synergistic effect of drug 1.

### 2.7. Production of HSPA5, HSPA6, and the Substrate-Binding Domain of HSPA5 and HSPA6

Full-length HSPA5 (26–636), HSPA6 (1–643), HSPA5 SBD (400–636), and HSPA6 SBD (396–643) were cloned into pSpeedET using ligation-independent cloning [33]. HSP70 constructs were expressed from pSpeedET vectors with a tobacco etch virus (TEV) protease cleavable N-terminal his tag. Codon plus *Escherichia coli* (Invitrogen) were transformed with the respective plasmid and grown on agar plates containing 35 μg/mL chloramphenicol and 50 μg/mL kanamycin. Colonies were washed into 2xYT media in baffled 2 L flasks and grown at 37 °C to an OD of 0.6. At this point, the bacteria were transferred to a room-temperature shaking incubator for 1 h before isopropyl β-d-1-thiogalactopyranoside (IPTG) was added to a final concentration of 500 μM. The bacteria were grown for 16 h and harvested by centrifugation. The pellets were resuspended in HKM buffer (50 mM HEPES pH 7.4, 150 mM KCl, 10 mM MgCl_2_, 2 mM β-mercaptoethanol (BME)) and lysed by repeated passage through a microfluidizer (microfluidics corporation) at 12,000 psi. The lysate was clarified by centrifugation and incubated with cobalt talon agarose from Gold Biotech (Olivette, MO, USA) for 1 h. The slurry was then applied to a gravity column and washed with 20 CV HKM buffer. The protein of interest was eluted with HKM buffer containing 200 mM imidazole. TEV protease was added to cleave the Hisx6 tag for 16 h at 4 °C. During this time, the eluted protein was extensively dialyzed against HKM buffer. Once cleavage was complete, the TEV protease was recaptured using cobalt resin and the protein was aliquoted and flash-frozen in liquid nitrogen.

### 2.8. Fluorescence Polarization Assay

FP assays were conducted in black 384-well low-volume plates (corning) with 10 nM of fluorescent peptide (FAM-ALLLSAPRR from ABclonal, Woburn, MA, USA) or JG-023 in assay buffer (50 mM HEPES pH 7.4, 100 mM KCl, 10 mM MgCl_2_, and 0.1% Triton X-100). Polarization was measured using the ID5 microplate reader (Molecular Devices, San Jose, CA, USA) with an excitation of 485 nm and emission of 535 nm after 4 h. For *K*_D_ determination, respective Hsp70s were serial diluted and the data were fit to a one-site binding model in PRISM 6.07 (GraphPad). For compound measurements, Hsp70 was added at a final concentration of 1 μM and % inhibition was calculated using unlabeled peptide as a positive control and DMSO as a negative control.

### 2.9. Determining Effect of CTCL Secreted Factors on Naïve T-Cell Activation via IFN-γ ELISA

IFN-γ levels in conditioned media were assessed via a human IFN-γ ELISA kit (Invitrogen Catalog No. 88-7316) Cells were treated as indicated, incubated at 37 °C 5% CO_2_ for 6 h, collected and spun down at 1000 rpm for 5 min, washed with RPMI 1640 with 2.05 mM L-Glutamine (Cytiva Catalog No. SH30027.01) not supplemented, and then resuspended in Immunocult-XF T-cell expansion media and incubated at 37 °C 5% CO_2_ for an additional 24 h. After 24 h, CTCL conditioned media were collected and spun down twice at 1500 rpm for 5 min to remove any remaining cells. PBMCs were plated with conditioned media and treated with human IL2 alone or in combination with Immunocult Human CD3/CD28/CD2 T-cell activator. Conditioned media alone were plated as a control. Plates were incubated at 37 °C 5% CO_2_ for 72 h. After 72 h, conditioned media were collected from PBMCs and IFN-γ levels were assessed via ELISA according to the manufacturer’s protocol. Results were analyzed in and compiled in GraphPad PRISM.

### 2.10. Cytokine Bead Array

Cells were stimulated with or without PMA and ionomycin for 6 h in the presence of drugs: JG-023 (10µM), Btz (5 nM), VER155008 (5µM), or combinations, followed by a wash-out. The cells were then incubated in fresh media for 24 h at 37 °C, 5% CO_2_. The conditioned media from stimulated and unstimulated cells for each of the treatment conditions were collected for human Th1/Th2/Th17 cytokine bead array (CBA) (BD Biosciences; Franklin Lakes, NJ, USA; Cat #560484). CBA samples and standards were prepared according to the manufacturer’s protocol and run on the LSR Fortessa flow cytometer (BD Biosciences; Franklin Lakes, NJ, USA). Samples were run undiluted or at 1:2 dilution in assay diluent. The data were analyzed using Flowjo software with the CBA plugin. Cytokine concentrations were calculated using the standard curve and values were normalized to cell counts.

## 3. Results

### 3.1. Divergent ER Stress Response Signaling in CTCL and Normal CD4+ T Cells

The role of UPR and ER stress signaling in the pathogenesis of T-cell malignancies, such as CTCL, is not known. To investigate this, we evaluated the effects of various pharmacological inducers of ER stress in a panel of CTCL cell lines and normal activated T cells. Cell models included HH cells, representative of the MF subtype, and Hut78, representative of the more aggressive SS form of CTCL. For normal T cells, we used activated PBMCs from healthy human donors and isolated CD4+ cells by immunomagnetic separation, as CD4+ is the dominant immunophenotype of CTCL [34,35] (Figure 1A). Specific ER stressors included drugs that inhibit vasolin-containing protein (VCP/p97) AAA ATPase (CB5083), Hsp90 (17-DMAAG), pan Hsp70 (VER155008), NEDD8-activating enzyme (NAE, MLN4924), HDAC6 (Ricolinostat), protein disulfide isomerase (PDI, LTI6426), and the 26S proteasome (bortezomib, Btz). These experiments showed that ER stress agents were widely cytotoxic to both CTCL and normal T cells (Figure 1B,C). They also revealed a statistically significant difference in sensitivity to Btz, with CTCL cells being >5-fold more sensitive than normal CD4+ in viability and apoptosis assays (Figure 1D,E). Upon investigation of signaling events in response to Btz, the difference between normal and malignant CD4+ T cells became even more evident. Btz induced a strong upregulation of classical UPR effectors such as ATF3, ATF4, CHOP, and HERPUD1 (Figure 1F). Normal CD4+ T cells, on the other hand, did not induce these pathways to any significant degree despite presenting with comparable levels of protein stress as indicated by the accumulation of high-molecular-weight poly-ubiquitinated protein smears (Figure 1F). Consistent with what was observed in viability and apoptosis assays, Btz induced apoptotic biomarkers such as the cleaved products of caspase substrates, PARP and Lamin A, in CTCL cells but not normal CD4+ T cells [36]. We confirmed that the observed UPR signaling was contributing to the apoptotic phenotype in experiments using the specific PERK inhibitor, GSK260614, which rescued HH and Hut78 cells from death induced by Btz (Appendix A). PERK is an apical stress sensor that triggers UPR signaling in response to ER stress. GSK260614 protected CTCL cells from Btz-induced death in a dose-dependent manner up to concentrations between 1 and 5 μM, concentrations above which it became cytotoxic as a single agent. Interestingly, PERK inhibition was unable to rescue cell death in normal primary T cells, suggesting different mechanisms of Btz-induced death between normal and malignant T cells. Together, these results reveal a divergence in the adaptive cellular response to ER stress induced by Btz between normal and malignant T cells that may indicate unique vulnerabilities of CTCL.

### 3.2. HSPA5/6 Play a Dominant Role in the ER Stress Response in CTCL Cells

To further characterize the UPR signaling profiles of malignant CTCL and normal CD4+ T cells, we used Btz as an ER stressor and evaluated the expression of a panel of UPR and oxidative stress genes that were selected previously [32]. Normal CD4+ cells showed a muted response to Btz with minimal activation of most gene targets in the set. CTCL, by comparison, showed a prominent activation of *ATF3*, the Hsp40 family member *DNAJB1*, and most notably, the inducible Hsp70 family member *HSPA6*, which was induced by 191- and 92-fold in CTCL cells versus only 5-fold in normal activated T cells (*p* < 0.001; Figure 2A). These hyper-responsive genes were further induced when Btz was combined with the PDI inhibitor LTI6426 (Appendix A), which is highly synergistic with Btz in CTCL (Appendix A) and a wide variety of other tumor types [31,37,38]. This suggests an important role of these genes and their products in CTCL’s response to ER in a variety of stress-inducing drugs. Given the pronounced induction of *HSPA6*, we next evaluated the other stress-inducible members of the Hsp70 family (*HSPA1A*, *HSPA1B*, *HSPA1L*, *HSPA5*, *HSPA6*) [39] in response to a panel of ER stressors. *HSPA6* showed the most prominent induction in response to Btz and several other drug classes compared to the other Hsp70 isoforms (Figure 2B). UPR-responsive genes were continuously induced over a 16 h time period, and again, HSPA6 showed the most significant induction (Figure 2C). Additionally, HSPA6 induction was consistently higher in CTCL cells than normal CD4+ T cells (Figure 2D). Results were confirmed at the protein level, as Western blotting analysis showed high HSPA6 induction in Btz-treated CTCL but not normal CD4+ cells (Figure 2E). A similar trend was observed for HSPA5/BiP/Grp78, although the difference between normal and malignant cells was less striking. Total Hsp70 levels were induced to similar degrees in both CTCL and normal CD4+ T cells (Figure 2E). This suggests that while cellular fitness and the signaling response to Btz-induced stress was very different between cell types, the magnitude of stress incurred by the cells was similar. This rules out potential variables that affect drug target engagement (e.g., drug efflux pumps) between cell types that could be responsible for these observations. Importantly, these findings reveal a dominant role for HSPA6 and HSPA5 in the CTCL adaptive response to ER stress-inducing drugs, leading us to hypothesize that these Hsp70 family members are attractive drug targets in CTCL.

### 3.3. HSPA5/6-Specific Inhibition Enhances ER Stress-Induced Death of CTCL Cells

Previous studies have identified Hsp70 isoform-selective inhibitors that target the substrate-binding pocket of the canonical human HSP70s with >100-fold selectivity for HSPA5 and HSPA6 compared to other isoforms [30]. Compound 35 from this series of analogs (herein referred to as JG-023; Figure 3A) showed selectivity for HSPA5 (also known as BiP or GRP78) and HSPA6. To further characterize the binding of JG-023 to HSPA5 and HSPA6, we generated a fluorescently labeled derivative, FAM-JG-023, using a copper-free click chemistry strategy (Schema in Figure 3B). We then titrated HSPA5 and HSPA6 against a fixed concentration (20 nM) of FAM-JG-023 in a fluorescence polarization (FP) assay to determine binding specificity and potency (Figure 3C). Competition binding assays were conducted using parent (unlabeled) JG-023 to displace FAM-JG-023 from full-length HSPA6 and HSPA5 and a truncated version of HSPA5 that encompasses the substrate-binding domain (SBD) mapping to amino acids 420–500 (Figure 3D). Unlabeled JG-023 successfully competed away FAM-JG-023 binding to HSPA6, HSPA5 FL, and HSPA5 SBD in a dose-dependent manner with IC50s of 0.5 ± 0.3, 1.3 ± 0.2, and 1.4 ± 0.06, respectively, confirming that JG-023 and FAM-JG-023 bound to the same binding pocket with specificity and good potency for an early-stage lead compound.

Given the hyper-responsiveness of the HSPA5 and A6 genes in the stress response of CTCL cells, we next evaluated the activity of JG-023 in our cell models. HSPA5/Bip/Grp78 is a chaperone that senses and regulates ER stress by engaging and inhibiting apical effectors of the UPR, including PERK, IRE1α, and ATF6 in non-stressed cells [40,41]. Misfolded proteins in the ER attract HSPA5 and sequester it from these effectors, thereby triggering their activation and UPR signaling. Because JG-023 inhibits HSPA5/6 function by binding to the SBD, we hypothesized that it would induce a UPR-like pseudo-response that mimics HSPA5/Bip/Grp78 displacement by misfolded proteins in the ER. Consistent with this, we observed the induction of key downstream UPR effectors, including ATF3, ATF4, and CHOP in a dose-dependent manner at the protein and mRNA levels upon treatment with JG-023 in CTCL cells (Figure 4A–C). HSPA6-specific inhibitors (Zxy-0028 and Zxy-0029) showed only modest activity with a slight induction of ATF4. In isobologram analyses, JG-023 showed synergistic killing when combined with Btz in CTCL cells, decreasing the EC50 by approximately 5-fold from 9.5 ± 2.3 to 1.9 ± 0.6 microM (Figure 4D). Similar synergy was observed with combinations of JG-023 and the NAE inhibitor MLN4924, and this was specific to CTCL cells and not normal CD4+ T cells (Appendix A). Similar synergies were detected with other commercially available non-selective HSP70 inhibitors, VER-155008 [42] and AP-4-139B [43], when combined with the second-generation proteasome inhibitor, carfilzomib (Figure 4E). These data suggest that HSP70 family members, including HSPA5 and A6, function in a protective role against ER stressors in CTCL cells. Furthermore, the combination of HSP70 inhibitors, such as JG-023, with pharmacological inducers of ER stress (e.g., proteasome and NAE inhibitors) is highly effective in killing CTCL cells.

### 3.4. HSPA5/6 Inhibition Suppresses Th2 Cytokine Production by CTCL Cells

Malignant T cells from advanced CTCL show a Th2-dominant phenotype, characterized by the secretion of IL-4, IL-5, IL-10, and IL-13 with low IFNγ production [14,44]. We evaluated the expression of a panel of Th1 and Th2 cytokine genes in our cell models and confirmed that HH and Hut78 CTCL cells showed a Th2 bias (*IL4*, *IL6*, *IL10*, and *IL13*) with relatively lower induction of Th1 cytokine genes (*IFNG*, *TNFA*, and *IL2)* compared to primary CD4+ T cells from normal healthy donors (Figure 5A). Similar results were observed following T-cell stimulation using phorbol 12-myristate 13-acetate (PMA) and ionomycin (Figure 5B). These transcriptional profiles for Th1 and Th2 cytokines were also confirmed at the protein level (Appendix A). CTCL also induced high levels of the *IL22* gene, an observation that has been reported in patients and shown to influence the makeup and architecture of the CTCL TME [45,46]. As previous studies have shown that Hsp70 regulates the expression of Th2 cytokines such as IL-10 in CD4+ T cells [47,48], we next investigated the impact of HSPA5/6 inhibition on the production of cytokines in CTCL cells. In normal CD4+ T cells, JG-023 significantly reduced Th2-type cytokines, including *IL4*, *IL5*, and *IL6*, along with *IL22* (Figure 5C). However, JG-023 had little effect on the expression of Th1-type cytokines (*IFNG*, *TNFA*, and *IL2)*. By comparison, in CTCL cells, JG-023 broadly suppressed the induction of cytokine genes with a prominent effect on Th2 cytokines including *IL4*, *IL10*, *IL13,* and *IL22* (Figure 5C). These results indicate that the inhibition of HSPA5 and HSPA6 suppresses the Th2 cytokine signature. Furthermore, in normal CD4+ T cells, JG-023 had no effect on Th1 cytokines, suggesting that HSPA5/6 may specifically regulate the Th2 differentiation program in normal T cells.

### 3.5. CTCL-Secreted Factors Suppress Normal T-Cell Function

Th2 cytokines play a key role in allergic responses and in defense and wound repair following helminth parasite infection [49,50]. They accomplish this in part by suppressing Th1 cytokines such as IFNγ and tipping the Th1/Th2 balance toward a Th2 bias [51]. Under normal conditions, Th2 immunity counterbalances Th1 immunity [52]. Under pathological conditions like CTCL, Th2 immunity has been proposed to induce global and local T-cell suppression in patients with CTCL [53]. Given the paucity of Th1 cells in CTCL lesions, we hypothesized that CTCL-derived secreted factors suppress normal Th1 cell activity. To test this, we activated naïve T cells from normal human donor PBMCs using a trimeric anti-CD3/CD28/CD2 antibody in the presence of conditioned media (CM) from HH and Hut78 CTCL cells. The activation of Th1-cell populations in PBMCs was measured using IFNγ production as a proxy. We found that CM from stimulated HH and Hut78 cells significantly repressed IFNγ production following the activation of normal T cells from three separate normal human donors in a dose-dependent manner (Figure 6A,B). Suppressive factors from CTCL cells exhibited high apparent potency with effects evident even when CM was diluted 50-fold (Figure 6B).

ER stress and the UPR play diverse roles in T-cell function, differentiation, and activation and TCR complex formation, as well as in T-cell pathologies [54,55,56]. Therefore, we next asked if the suppressive factors secreted by CTCL cells were dependent on HSP70 family members including HSPA5/6. To test this, we collected CM from activated CTCL cells that had been treated with JG-023 or the pan HSP70 inhibitor VER155008. We also tested the effects of Btz alone and in combination with HSP70 inhibitors given the synergy we observed between these two classes of drugs in prior experiments (Figure 4D,E). CTCL cells were activated with PMA + ionomycin for six hours in the presence and absence of drug. They were then washed to remove PMA/ionomycin and drug and incubated in basic T-cell media for an additional 24 h, after which, CM were collected. CM volumes were adjusted to viable cell count to normalize for any effects of the drugs on cell viability. Finally, normal T cells from human PBMCs were activated in the presence of CM from the different treatment groups to determine whether HSP70 inhibitors and ER stressors like Btz could block the immunosuppressive effects of CTCL. Indeed, we found that JG-023 monotherapy and the combination of JG-023 + Btz were able to rescue IFN-γ levels in normal T cells, indicating a restoration of normal Th1 cell activity (Figure 6C). VER155008 showed a modest but statistically insignificant rescue of IFN-γ as a single agent and did not add significantly to the activity of Btz alone.

To evaluate the contribution of select cytokines to this suppressive effect, we next quantified the concentrations of Th2 and Th1 cytokines in the CM from treated and untreated CTCL cells. Cytokine bead array analysis revealed that JG-023 and the combination of JG-023 and Btz reduced IL-10 levels in both resting and activated CTCL cells (Figure 6D). Similar results were observed for IFN-γ, IL-2, and to a lesser extent, TNFα, whereas minimal effects were observed for IL-4, IL-6, and IL-17a. The pan HSP70 inhibitor, VER155008, alone and in combination with Btz, had similar effects on Th1 cytokines, IFN-γ, and IL-2, but had no effect on IL-10 levels in either resting or activated CTCL cells. As JG-023 but not VER155008 blocked IL-10 production by CTCL cells and restored normal Th1 cell activation, we hypothesized that IL-10 was a key factor in the immunosuppressive phenotype of CTCL cells. To test this, we activated normal T cells in the presence of suppressive CM from CTCL cells and co-treated with neutralizing antibodies to IL-4, IL-6, IL-10, and IL-13. We found that only the IL-10-neutralizing antibody rescued normal T-cell activation (Figure 6E), indicating that IL-10 was indeed critical, at least in part, for the T-cell suppressing activity of CTCL cells. In summary, we show that CTCL-derived cytokines suppress normal Th1 cell activation. This repressive activity can be overcome by drugs that modulate the ER stress response, such as Btz-, pan HSP70-, and HSPA5/6-specific inhibitors. Lastly, IL-10 is a major inhibitory factor released by CTCL that contributes to the suppression of Th1 cells. Altogether, these data suggest that pharmacological interventions targeting the ER stress pathway, especially those that target HSPA5/6, not only induce anti-CTCL effects that include a strong stress and apoptotic signaling response but also inhibit their immunosuppressive Th2 phenotype. 

## 4. Discussion

The ER stress response has been linked to various aspects of cancer initiation and progression, including cell growth and protection from apoptosis [57,58,59]. In the current study, we demonstrate a divergence in the ER stress response between CTCL and normal CD4+ T cells. Specifically, we demonstrate that CTCL cells are hyper-responsive to ER stress-inducing drugs (e.g., proteasome, NAE, and PDI inhibitors), exhibit the heightened activation of UPR effectors like HSPA6 in response to ER stressors, and present a Th2-suppressive phenotype bias that can be altered using inhibitors of specific UPR pathway components. This may suggest an important role of ER stress in CTCL disease initiation, progression, and T-cell biology and a new path for the treatment of CTCL in the clinic.

Our work highlights the significance of specific HSP70 isoforms, HSPA5 (BiP/Grp78) and HSPA6, in the CTCL immune phenotype and response to ER stress-inducing drugs. HSPA5, a glucose-regulated protein first identified in 1974 [60], is widely recognized for its overexpression in a variety of cancers and has emerged as a prognostic biomarker of disease and predictor of infiltrating immune cells in the tumor microenvironment [61,62,63]. By comparison, much less is known about HSPA6, which is at least partially due to it not being conserved in rodents [64,65]. HSPA6 (also known as HSP70B’) was initially sequenced and characterized as a heat- and stress-inducible HSP70 family member [65] and later described for its proinflammatory role in atherosclerotic plaque formation [66,67]. HSPA6 expression has been reported in certain tumor types and associated with outcomes, although its precise role in cancer is not well understood. Moreover, little is known about the expression and role of both HSPA5 and HSPA6 in CTCL. Our study highlights the unique role of HSPA5 and HSPA6 proteins in the ER stress response of CTCL, but not of normal CD4+ T cells. We observed the significant induction of HSPA5 and HSPA6 genes as well as increased protein expression in the ER stress response following treatment with Btz. We also observed enhanced ER stress-induced apoptosis in CTCL cells with the HSPA5/6-selective, inhibitor JG-023. Our findings are consistent with a recent study which showed Hsp70 to be upregulated in response to Btz as well as a synergistic relationship between Hsp70 inhibition and Btz [68]. The differential expression of HSPA5 and HSPA6 between CTCL and normal CD4+ T cells highlights an important role of ER stress signaling and the UPR in malignant T-cell biology and identifies these proteins as potential therapeutic targets for CTCL. Additionally, while Btz treatment has been utilized and proven effective as a single-agent therapeutic in CTCL, acquired resistance is a known barrier to its efficacy, and recent studies have linked the higher expression of heat shock proteins with acquired resistance to Btz [69,70]. Therefore, the induction of HSPA5 and HSPA6 in response to Btz in CTCL may play a cytoprotective role, and prolonged exposure and targeting HSPs such as HSPA5 and HSPA6 could enhance the efficacy of ER stress-inducing compounds like Btz. It is also important to note that pro-apoptotic ER stress signaling is a major component of the anti-cancer mechanism of Btz, but it is not the only one, as proteasome inhibition is known to have pleiotropic effects in cancer cells [71]. Our results show the clear convergence of HSPA5/6 inhibition and Btz on ER stress signaling, but we cannot rule out the possibility that HSPA5/6 inhibitors and Btz also synergize on non-ER stress pathways as well.

The ER stress response and UPR signaling pathways play a role in CD4+ T helper cell development, differentiation, and effector function [54,55,72]. In normal T-cell development, evidence supports a critical role for UPR effectors such as the chaperone gp96 (HSP90b1, grp94) in the thymus [73], while other HSP families, such as members of the heat shock protein 70 (HSP70) family, play a controversial and potentially stage-dependent role in T-cell differentiation and activation. For example, inducible HSP70 isoforms promote T-cell activity in the context of autoimmunity, whereas other reports have shown that HSP70 inhibits T-cell development early in thymic development [74,75,76]. In the current study, we demonstrate the Th2 skewed phenotype of HH and Hut78 CTCL could be normalized with selective HSPA5/6 inhibition. Interestingly, HSPA5/6 inhibition in normal CD4+ T cells suppressed Th2 cytokine production but not Th1 cytokines, suggesting a dominant role for HSPA5/6 in signaling in Th2 cell function and differentiation in both malignant and normal CD4+ T cells. While several studies have demonstrated the role of UPR signaling members such as IRE1α/XBP1 as important for promoting Th2 differentiation in normal T cells [48,49,77], less is known about the role of HSPA5 and HSPA6 in the function of CD4+ Th2 cells, specifically, or broadly in the overall differentiation and function of T-cell subsets. One of the novel findings of this study is the spotlighting of HSPA5 and HSPA6 in the stress response of CTCL and the Th2 phenotype. Further studies are required to better understand this role, and the potential therapeutic implications that our work suggests warrant additional mechanistic studies and the optimization of HSPA5/6 isoform-selective inhibitors.

The Th2 immune phenotype of CTCL manifests as predominantly immunosuppressive [53,78,79]. By comparison, under normal conditions, Th2 immunity mediates a proinflammatory response against environmental allergens and helminth parasites and accelerates wound healing and damage repair at sites of infection [80,81,82]. The expansion of Th2-dominant malignant T cells in CTCL is thought to disrupt the balance between Th1 and Th2 immunity and favor a Th2-dominant TME which is immunosuppressive and resistant to immune surveillance and tumor control. Our findings support this as we found that CTCL-derived secreted factors suppressed the activation of normal naïve T cells. We were able to partially overcome the suppression using HSAP5/6 selective inhibitors and Btz, which we also showed could block Th2 cytokine production, demonstrating causality between ER stress and effectors of the UPR and the immunosuppressive phenotype of CTCL. Of the primary Th2 cytokines (i.e., IL-4, IL-5, IL-10, IL-13), we identified IL-10 as the dominant driver of CTCL-mediated immunosuppression, as we were able to rescue IFN-γ production in normal T cells with the addition of an IL-10-neutralizing antibody. As systemic immunosuppression and susceptibility is the leading cause of death amongst CTCL patients, identifying and targeting factors influencing immunosuppression is important for improving current standard of care and patient outcomes. While therapies exist that have been reported to decrease Th2 cytokine production in CTCL patients, such as bexarotene and IFN-α, resistance to these therapies has been reported, and no increase in overall survival has been observed [78]. Our work suggests that other targeted strategies, such as HSPA5/6-selective inhibitors alone or in combination with Btz, should be explored further for their potential to both overcome the immunosuppressive nature of CTCL and improve overall patient survival which the current standard of care is unable to do.

Future studies should further explore the potential of small-molecule inhibitors of HSPA5/6 for the treatment of CTCL in vivo and primary CTCL patient cells. Other methods of inhibition should also be explored, including the use of HSPA5/6-specific proteolysis targeting chimeras (PROTACs), which target proteins for proteasomal degradation and do not rely on the inhibition of target enzymatic function. Additional studies are also required to better understand the contribution of CTCL-derived secreted factors to the immune microenvironment and effects on global immune function. CTCL may affect extensive portions of the total body surface area (BSA), ranging anywhere from <10% of BSA in stage IA, to 10–25% of BSA in stage IB/IIA, and >25% of BSA in stage IB/IIA and more advanced disease forms [83]. The extent of disease burden coupled with the secretion of Th2 cytokines is likely to polarize the immune system away from Th1 immunity and have a significant impact on the ability of the host to fight infections. In fact, reports confirm that CTCL indeed suppresses Th1 responses, thereby reducing anti-tumor immunity in CTCL skin lesions and decreasing systemic immunity against microbial pathogens [20,84]. In addition, further evidence suggests that rebalancing toward the Th1 response correlates with a positive response to therapy in CTCL patients [53]. Our results suggest that therapeutic strategies to restore the Th1/Th2 immune balance should be explored in CTCL, and existing UPR-inducing drugs, like Btz, or novel HSPA5/6-selective inhibitors are potential candidates for pursuing that objective.

## 5. Conclusions

We identified divergent UPR signaling between malignant CTCL and normal CD4+ T cells that involves the hyper-responsiveness of Hsp70 family members, HSPA5 and HSPA6. These findings implicate HSA5/6 in CTCL biology, suggest a unique sensitivity of CTCL to proteasome inhibitors and isoform-selective HSPA5/6 inhibitors (JG-023), and connect Hsp70 and UPR signaling to the Th2-dominant immunosuppressive phenotype of CTCL. 

## Figures and Tables

**Figure 1 biomolecules-15-00076-f001:**
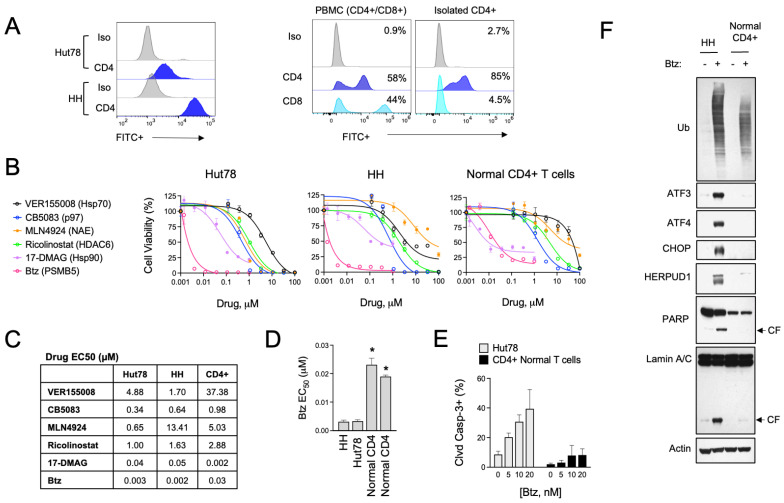
Btz selectively targets CTCL cells over normal CD4+ T cells. (**A**) (**Left**) Cell surface CD4 expression was measured by flow cytometry in Hut78 and HH CTCL cells. (**Right***)* Non-malignant CD4+ T cells were isolated from PBMCs from normal human donors, which are a mixture of CD4+ and CD8+ T-cell populations. Flow cytometry data are shown for CD8- and CD4-stained cells. (**B**) Hut78, HH, and normal CD4+ T cells were treated with dose ranges of the indicated ER stress-inducing drugs for 48 h. Cell viability data are shown. (**C**) Effective concentration 50 (EC_50_) values were extrapolated from dose curves shown in (**B**). EC50 values are shown in μM. (**D**) Effective concentration 50 (EC_50_) values were extrapolated from Btz dose curves representing 2–3 experiments for each of the indicated cell lines. CD4+ T cells from 2 different human donors are shown. Statistical significance was determined using Student’s *t*-test (* *p* < 0.05, N = 3). (**E**) Hut78 CTCL cells and normal CD4+ T cells were treated with the indicated concentrations of Btz for 24 h. Flow cytometry data for cleaved caspase-3 positive cells are shown (mean ± SEM, N = 3). (**F**) HH and normal CD4+ T cells were treated with Btz (20 nM) for 16 h. Western blot analysis of the indicated UPR and apoptotic markers is shown.

**Figure 2 biomolecules-15-00076-f002:**
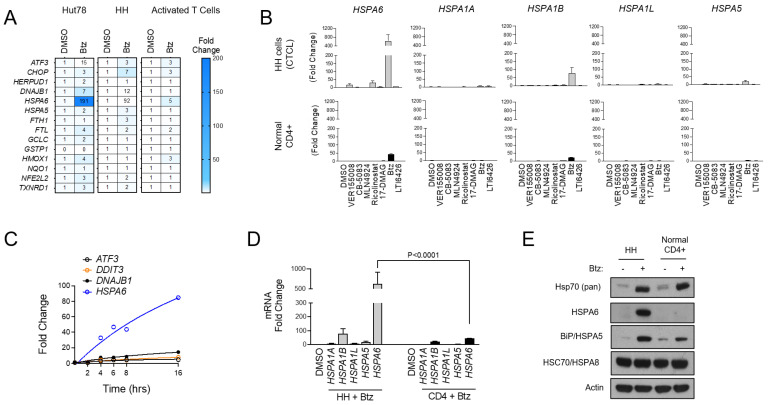
HSPA6 is a biomarker of response to Btz in CTCL cells. (**A**) Hut78, HH, and normal CD4+ T cells were treated with Btz for 16 h. RT-qPCR using primers for the indicated transcripts was conducted. Data were normalized to *GAPDH* (internal control) and DMSO-treated (treatment control) groups. (**B**) HH and normal CD4+ T cells were treated with an EC_50_ dose of the indicated ER stress-inducing drugs. RT-qPCR data for the indicated HSP70 isoform gene transcripts are shown. Data are expressed as average fold change ± SEM after normalization to *GAPDH* (internal control) and DMSO (treatment control). (**C**) HH cells were treated with Btz (10 nM) and expression levels of the indicated UPR genes were measured over time via RT-qPCR. Data are presented as fold change. (**D**) HH and normal CD4+ T cells were treated with Btz (20 nM) for 16 h. RT-qPCR data using primers for the indicated HSP70 isoform transcripts are shown. Statistical significance was determined using Student’s *t*-test (N = 3). (**E**) HH and normal CD4+ T cells were treated with Btz (20 nM) for 24 h. Western blots for the indicated HSP70 proteins are shown.

**Figure 3 biomolecules-15-00076-f003:**
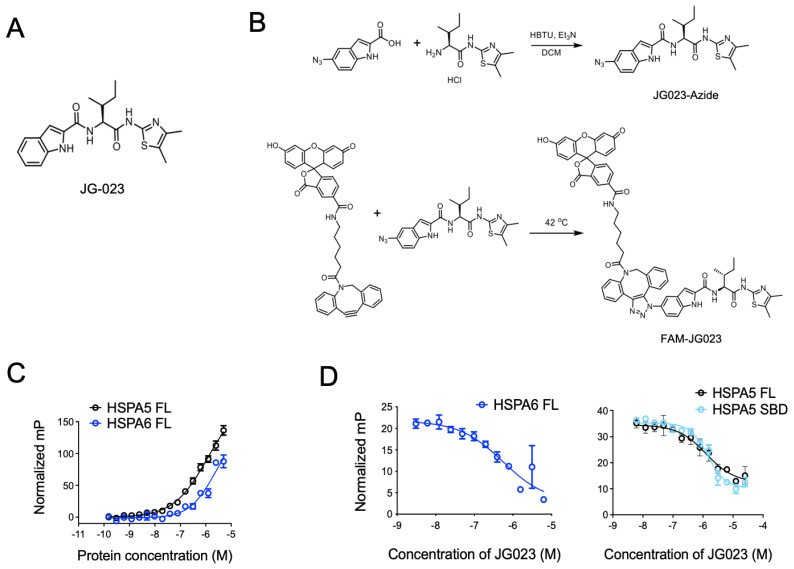
Selective HSPA5/6 inhibition using compound JG-023. (**A**) The chemical structure of JG-023 is shown. (**B**) A fluorescently labeled JG-023 (FAM-JG023) analog to be used in fluorescence polarization studies was synthesized using a copper-free click chemistry strategy. Chemical synthesis schema is shown. (**C**) Full-length (FL) HSPA5 and HSPA6 were titrated against 20 nM FAM-JG023. Fluorescence polarization binding data are shown. (**D**) Competition experiments were conducted using 1 mM HSPA6 FL, HSPA5 FL, and the HSPA5 substrate-binding domain (SBD) and 20 nM FAM-JG023. A dose range of unlabeled JG-023 was used to compete off FAM-JG023. Binding data from a fluorescence polarization assay are shown.

**Figure 4 biomolecules-15-00076-f004:**
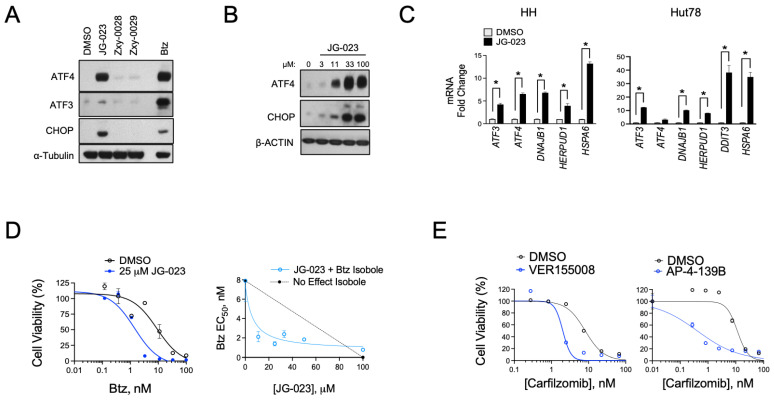
HSPA5/6-selective inhibitors induce the UPR and enhance proteasome inhibitor sensitivity in CTCL models. (**A**) Hut78 cells were treated with the indicated HSPA5/6- (JG-023) or HSPA6- (Zxy-0028 and Zxy-0029) selective inhibitors at 20 μM for 16 h. Western blots are shown. (**B**) HH cells were treated with a dose range of JG-023 for 16 h. Western blots are shown. (**C**) HH and Hut78 cells were treated with DMSO (control) or JG-023 (10 μM) for 16 h. RT-qPCR data are shown normalized to GAPDH (internal control) and DMSO (treatment control). Statistical significance was determined using Student’s *t*-test (* *p* < 0.05, N = 3). (**D**) (**Left**) HH cells were treated with a dose range of Btz in the presence or absence (DMSO) of JG-023 (25 μM). Cell viability data are shown (DMSO: EC50 = 9.5 ± 2.3 μM; JG-023: EC50 = 1.9 ± 0.6 μM). (**Right**) Isobologram analysis is shown and confirms synergy between Btz and JG-023. The dotted line represents the no-effect isobole. (**E**) Hut78 cells were treated with a dose range of the second-generation proteasome inhibitor, carfilzomib, in the absence (DMSO) and presence of the pan HSP70 inhibitors, VER155008 (**Left**, 5 μM) and AP-4-139B (**Right**, 5 μM). Cell viability data are shown (carfilzomib EC50: DMSO = 8.1 ± 2.5 μM; VER155008 = 2.0 ± 0.6 μM; DMSO = 11.4 ± 3.3 μM; AP-4-139B = 0.42 ± 0.2 μM).

**Figure 5 biomolecules-15-00076-f005:**
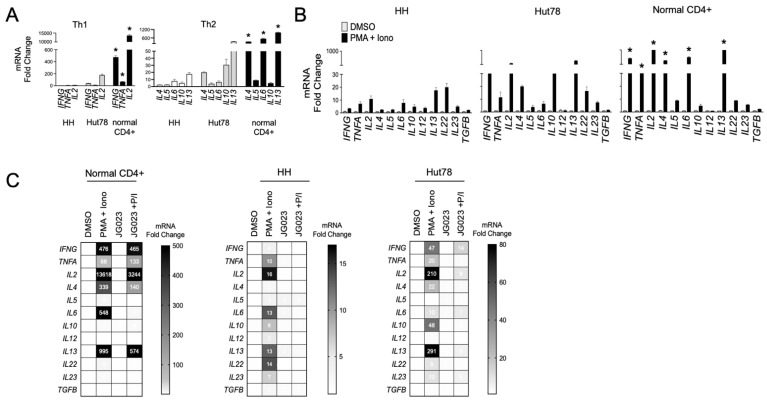
HSPA5/6 inhibition represses the Th2 phenotype in CTCL cells. (**A**) RT-qPCR was conducted on untreated HH, Hut78, and normal CD4+ T cells using primers targeting the indicated gene transcripts. Data were normalized to GAPDH to generate a relative fold change (mean ± SEM, N = 3). * *p* < 0.01 comparing normal CD4+ to both HH and Hut78 using a one-way ANOVA. (**B**) HH, Hut78, and normal CD4+ T cells were activated using phorbol ester (PMA, 1 μM) and ionomycin (Iono, 200 ng/mL) for 16 h and mRNA levels for the indicated cytokine gene panels were analyzed. RT-qPCR data are shown normalized to GAPDH (internal control) and DMSO (treatment control) and expressed as fold change (mean ± SEM, N = 3). * *p* < 0.01 comparing normal CD4+ to both HH and Hut78 using a one-way ANOVA. (**C**) The indicated T cells were activated with PMA + Iono in the presence or absence of JG-023 (20 μM) for 6 h. Cells were washed to remove treatments and incubated in fresh media for an additional 18 h. mRNA transcript levels for the indicated gene targets were then quantified. RT-qPCR data are shown in heat map format. Data were normalized to GAPDH (internal control) and DMSO (treatment control).

**Figure 6 biomolecules-15-00076-f006:**
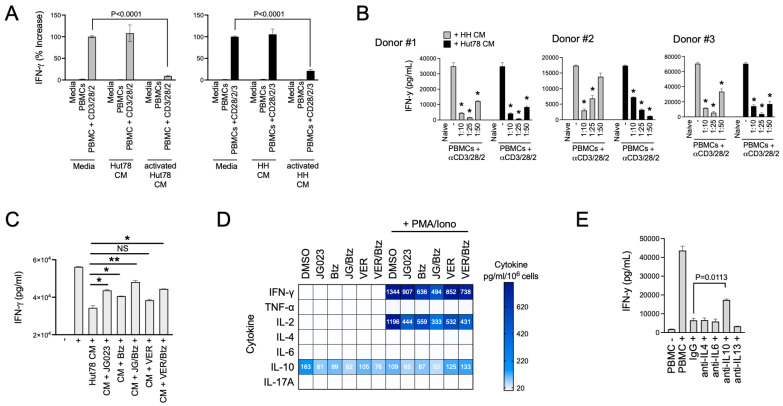
CTCL cells suppress normal T-cell activation. (**A**) Normal T cells from healthy human PBMCs were activated in the presence of conditioned media (CM) from resting and activated Hut78 and HH cells. IFNγ levels were determined by ELISA (mean ± SEM, N = 3). Normal T cells were activated using a trimeric anti-CD3/28/2 antibody. CM from activated CTCL cells were collected by treating Hut78 and HH cells with a combination of PMA + ionomycin for 6 h, after which, cells were washed to remove drug, and cells were finally incubated in fresh media for an additional 18 h. Statistical significance was determined using Student’s *t*-test (N = 3). (**B**) Experiments described in (**A**) were conducted using normal T cells from 3 healthy human donors and varying dilutions of CM from Hut78 and HH cells. * *p* < 0.0001 compared to T-cell stimulation alone (“-”) by Student’s *t*-test (N = 3). (**C**) Normal T cells from healthy human donors were activated (+) in the absence or presence of Hut78 CM that were collected under the indicated conditions of treatment with JG-023 (10 μM), Btz (5 nM), VER155008 (5 mM), or combinations thereof. Hut78 cells were activated with PMA and ionomycin for 6 h in the presence of drugs followed by a wash-out. IFNg levels in the media from normal T-cell activation cultures were measured by ELISA after 48 h (mean ± SEM, N = 3). Statistical significance was determined by Student’s *t*-test (* *p* < 0.05, ** *p* < 0.01, NS = no statistical significance). (**D**) Hut78 CM from the treatment groups described in (**A**) were analyzed by cytokine bead array for the indicated cytokines. To account for any effects of the drugs on cell viability and proliferation, cytokine levels were normalized to the number of viable cells counted at the time of CM collection. Data are expressed as pg/mL per 10^6^ viable Hut78 cells. (**E**) Normal T cells from healthy human donors were activated in the presence of Hut78 CM and neutralizing antibodies to the indicated cytokines. IFNg levels in the media from activated normal T-cell cultures were analyzed by ELISA (mean ± SEM, N = 3). Statistical significance was determined using Student’s *t*-test.

**Table 1 biomolecules-15-00076-t001:** Primary and secondary antibodies.

Primary Antibodies		
Protein Target	Vendor	Catalog #
β-actin	Sigma	A5441
ATF3	Cell Signaling	33593S
ATF4	Cell Signaling	11815S
BiP	Cell Signaling	3183S
CHOP	Cell Signaling	2895S
eIF2α	Cell Signaling	5324T
p-eIF2α	Cell Signaling	3398S
HERPUD1	Cell Signaling	26730
HSC70/HSPA8	Santa Cruz	71270
HSPA6	Santa Cruz	374589
HSP70	Cell Signaling	4872S
Tubulin	Cell Signaling	3873S
Ubiquitin	Cell Signaling	3936S
XBP1S	Cell Signaling	12782S
**Secondary Antibodies**	**Vendor**	**Catalog #**
Goat anti-Mouse IgG-H + L	Invitrogen	31430
Goat anti-Rabbit IgG-H + L	Invitrogen	31480
Goat anti-Rat IgG-H + L	Invitrogen	9520

**Table 2 biomolecules-15-00076-t002:** RT-qPCR primer sequences.

Target	Fwd (5′-3′)	Rvs (5′-3′)
ATF3	GGAGTGCCTGCAGAAAGAGT	CCATTCTGAGCCCGG ACAAT
ATF4	GACGGAGCGCTTTCCTCTT	TCCACAAAATGGACGCTCAC
DDIT3	GGAAACAGAGTGGTCATTCCC	CTGCTTGAGCCGTTCATTCTC
DNAJB1	CCAGTCACCCACGACCTTC	CCCTTCTTCACTTCGATGGTCA
HERPUD1	CCGGTTACACACCCTATGGG	TGAGGAGCAGCATTCTGATTG
HSPA1A	GGCCTTGAGGACTTTGGGTTA	TGGGAATGCAAAGCACACG
HSPA1B	GGGAGGACTTCGACAACAGG	GACAAGGTTCTCTTGGCCCG
HSPA1L	AAAGCAGGTCAGGGAGAGCGA	GGAGGGATTCCAGTCAGGTCA
HSPA5	GGGAGGTGTCATGACCAAAC	GCAGGAGGAATTCCAGTCAG
HSPA6	GATGTGTCGGTTCTCTCCATTG	CTTCCATGAAGTGGTTCACGA
IL2	AGAATCCCAAACTCACCAGGATGC	AGATGTTTCAGTTCTGTGGCCTTC
IL4	ACAGCCTCACAGAGCAGAAGAC	TCTCATGGTGGCTGTAGAACTGC
IL5	GGCACTGCTTTCTACTCATCGA	AGTTGGTGATTTTTATGTACAGGAACA
IL6	TCTCCACAAGCGCCTTCG	CTCAGGGCTGAGATGCCG
IL10	GCTGGAGGACTTTAAGGGTTACCT	CTTGATGTCTGGGTCTTGGTTCT
IL12b	TCATCAAACCTGACCCACCCAAGA	TTTCTCTCTTGCTCTTGCCCTGGA
IL13	GAAGGCTCCGCTCTGCAAT	TCTGGGTCTTCTCGATGGCA
IL22	GCAGGCTTGACAAGTCCAACT	GCCTCCTTAGCCAGCATGAA
IL23a	ACTCAGCAGATTCCAAGCCTCAGT	TGGAGATCTGAGTGCCATCCTTGA
IFNg	TCCAAGTGATGGCTGAACTGTCG	ACCTCGAAACAGCATCTGACTCC
TGFb1	CAAGCAGAGTACACACAGCAT	TGCTCCACTTTTAACTTGAGCC
TNFa	CCAGGCAGTCAGATCATCTTCTCG	ATCTCTCAGCTCCACGCCATTG

## Data Availability

The original contributions presented in this study are included in the article. Further inquiries can be directed to the corresponding author.

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
