# Peer review of "Pharmacological Modulation of the Unfolded Protein Response as a Therapeutic Approach in Cutaneous T-Cell Lymphoma"

_biomolecules, 2025, doi:10.3390/biom15010076_

Round 1

Reviewer 1 Report

Comments and Suggestions for Authors

The authors explore the role of unfolded protein response pathway in cutaneous T cell lymphoma, specifically HSP70 family protein: HSPA5/6. Interestingly, Btz selectively induces UPR and apoptosis in HH, but not in normal CD4+ cells. Moreover, they found a compound JG-023 which synergically increases toxicity of UPR inducers in CTCL. The manuscript is well organized and comprehensively described. The data analysis and conclusions look reasonable

Additionally, here are some recommendations for refinement.

1.           there is no analysis part in material and methods. In addition, the authors didn’t mention the analysis methods for RT-qPCR. Did the authors use ΔΔCt methods.

2.           The authors mentioned a couple of times of RT-PCR: 459 ,454 and 422 lines. Was it a typo for RT-qPCR?

3. The authors should consider add analysis results in Figure 4D&E

4. The authors mentioned that they used student t test for Figure 6C, which includes multiple groups. Did you consider using ANOVA analysis to analyze multiple groups data?

Author Response

Reviewer #1

The authors explore the role of unfolded protein response pathway in cutaneous T cell lymphoma, specifically HSP70 family protein: HSPA5/6. Interestingly, Btz selectively induces UPR and apoptosis in HH, but not in normal CD4+ cells. Moreover, they found a compound JG-023 which synergically increases toxicity of UPR inducers in CTCL. The manuscript is well organized and comprehensively described. The data analysis and conclusions look reasonable

Additionally, here are some recommendations for refinement.

Thank you for your positive feedback and thoughtful comments. We have addressed your critiques as described below.

In addition, the authors didn’t mention the analysis methods for RT-qPCR. Did the authors use ΔΔCt methods.

Yes, we used the ΔΔCt method. This has been clarified in the Materials and Methods section on page 11, lines 212-214.

The authors mentioned a couple of times of RT-PCR: 459 ,454 and 422 lines. Was it a typo for RT-qPCR?

Yes, this was a typo and the corrections have been made in the text.

The authors should consider add analysis results in Figure 4D&E

We have added the EC50 +/- SEM values for Btz in the presence or absence of JG-023 to the Figure legend for Figure 4D and EC50 +/- SEM values for Carfilzomib in the presence or absence of VER155008 and AP-4-139B.

The authors mentioned that they used student t test for Figure 6C, which includes multiple groups. Did you consider using ANOVA analysis to analyze multiple groups data?

We chose the student’s t-test for this statistical analysis because we were comparing the means of two groups – the T cell activation (IFNg response) in the presence of Hut78 CM versus in the presence of the indicated drugs. Our goal was to identify the drugs/combinations that restored the IFNg T cell response. The ANOVA analysis would have been appropriate if we were comparing all groups in the set, but that was not our objective.

Reviewer 2 Report

Comments and Suggestions for Authors

Dr. Dolloff explores how CTCL cells respond to pharmacological inducers of UPR and compare with normal T-cells. CTCL cells are more sensitive to e.g., bortezomib than normal CD4+ T cells. CTCL cells exhibit heightened activation of UPR markers like HSPA5 and HSPA6, while normal T cells show minimal induction of these markers. In addition, this study highlights the therapeutic promise of targeting UPR pathways in CTCL, HSPA5/6 inhibition, to induce apoptosis in malignant cells and counteract their immunosuppressive effects. However, this study lacks RNA-seq data to comprehensively assess global gene expression changes between treatment and untreated conditions. This limits insight into the broader transcriptional impact of UPR-modulating therapies. And I am curious how gene expression changes across multiple treatment time points, which could provide a deeper understanding of the dynamics of UPR signaling and therapeutic response. Despite these limitations, the study is still highly informative to the research community, providing valuable insights into UPR signaling and potential therapeutic interventions in CTCL.

Author Response

Reviewer #2

Dr. Dolloff explores how CTCL cells respond to pharmacological inducers of UPR and compare with normal T-cells. CTCL cells are more sensitive to e.g., bortezomib than normal CD4+ T cells. CTCL cells exhibit heightened activation of UPR markers like HSPA5 and HSPA6, while normal T cells show minimal induction of these markers. In addition, this study highlights the therapeutic promise of targeting UPR pathways in CTCL, HSPA5/6 inhibition, to induce apoptosis in malignant cells and counteract their immunosuppressive effects.

However, this study lacks RNA-seq data to comprehensively assess global gene expression changes between treatment and untreated conditions. This limits insight into the broader transcriptional impact of UPR-modulating therapies.

And I am curious how gene expression changes across multiple treatment time points, which could provide a deeper understanding of the dynamics of UPR signaling and therapeutic response. Despite these limitations, the study is still highly informative to the research community, providing valuable insights into UPR signaling and potential therapeutic interventions in CTCL.

Agreed, RNA-seq would provide a more comprehensive overview of global gene expression changes and insight into the broader transcriptional impact of UPR-modulating drugs. We conducted RNA-Seq analysis in a previous study (Duncan et al. Cancer Res 2020) which identified the candidate gene set (Fig. 2A) that we focused on in this study.

With regards to gene expression changes over treatment time, we performed an experiment evaluating major UPR signaling genes over a 16hr time course with Bortezomib in HH cells. We observed a time dependent and continuous induction of HSPA6, DDIT3/CHOP, DNAJB1, and ATF3. Consistent with our other findings, HSPA6 was the most responsive gene in the set. This further emphasizes that HSPA6 plays a critical role in the UPR signaling response of CTCL cells and highlights its potential as a therapeutic target. Data from this experiment is now included as part of Figure 2 (See new Fig 2C). The figure panel and manuscript text on line 342 have been revised to include this update.   

Reviewer 3 Report

Comments and Suggestions for Authors

The manuscript by St Thomas et al. “Pharmacological modulation of the Unfolded Protein Response as a therapeutic approach in cutaneous T cell lymphoma” describes consequences of endoplasmic reticulum stress induction on the viability and production of cytokines by the cutaneous T cell lymphoma cells. Authors revealed HSPA6 and HSPA5 as putative targets for the combined therapy with ER-stress inducers and proteasome inhibitors. Generally, I liked the manuscript, it is well structured, clearly written and contains interesting data. I suppose that it can be published in Biomolecules after the minor revision.

Please find my comments below:

Although selected drugs were shown to induce the ER-stress, several important compounds are missing in the list including tunicamycin, thapsigargin, and Brefedin A, as well as the physiological ER-stress inducers [PMID: 21266244]. Therefore, I would kindly ask authors to comment on the selection of the ER-stressors.

Bortezomib inhibits 20S proteasome catalytic activity and hence blocks the ER-associated protein degradation (ERAD) which in turn causes misfolded proteins to accumulate in the ER leading to the ER stress. At the same time, proteasome blockade affects degradation of hundreds of different proteins both in the cytoplasm and in the nucleus. Thus, I assume that a care should be taken when proteasome inhibitors are considered as specific ER-stress inducers;

I assume that exact IC50 values obtained following the incubation of compounds with CTCL cells should be indicated in the main text;

I would recommend putting the sequences of primers and antibodies into Tables;

Please describe the acquisition of HSPA6 FL, HSPA5 FL, and the HSPA5 substrate binding domain (SBD) in Materials and methods;

Please describe fluorescence polarization (FP) assay in Materials and methods;

Was the sole effect of JG-023 on cell viability addressed?

Perhaps, authors can benefit from using SynergyFinder 3.0 software (https://synergyfinder.fimm.fi);

Line 369 “with IC50s of 0.5±0.3, 1.3±0.2, and 1.4±0.06, respectively” please indicate the concentration;

Fig.1B several error bars are missing;

Fig.5 A,B, please indicate statistically significant changes;

Fig.5 C Please correct CD4+ to Normal CD4+;

Line 443-444 “In normal CD4+ T cells, JG-023 significantly reduced Th2 type cytokines, including IL4, IL5, IL6, along with IL22” however, there is no visible difference in IL5 and IL22 gene expression in normal CD4+ cells (Fig. 5C). From the Figure 5C one can deduce that JG023 suppresses both Th1 and Th2 cytokine expression in CTCL cells;

Fig. 6 B, D, please indicate statistically significant changes. Please comment on zero levels of IFNg or IL-2 in Hut78 CM despite certain mRNA content shown in Figure 5A. I suggest modification of Fig6. D;

Line 476-478 “We found that CM from stimulated HH and Hut78 cells significantly repressed IFNg production following activation of normal T cells from three separate normal human donors in a dose dependent manner (Fig. 6A-B)” Were the PBMCs used in the experiment shown in Figure 6A also obtained from three different donors?

Author Response

Reviewer #3

The manuscript by St Thomas et al. “Pharmacological modulation of the Unfolded Protein Response as a therapeutic approach in cutaneous T cell lymphoma” describes consequences of endoplasmic reticulum stress induction on the viability and production of cytokines by the cutaneous T cell lymphoma cells. Authors revealed HSPA6 and HSPA5 as putative targets for the combined therapy with ER-stress inducers and proteasome inhibitors. Generally, I liked the manuscript, it is well structured, clearly written and contains interesting data. I suppose that it can be published in Biomolecules after the minor revision.

Please find my comments below:

Although selected drugs were shown to induce the ER-stress, several important compounds are missing in the list including tunicamycin, thapsigargin, and Brefedin A, as well as the physiological ER-stress inducers [PMID: 21266244]. Therefore, I would kindly ask authors to comment on the selection of the ER-stressors.

Agreed, there are a number of ER-stress inducing compounds that we could have studied. We chose ones with the highest clinical relevance as defined by those that have been approved by the FDA or those that previously or are currently in clinical trials in man. We reasoned that these molecules had the most translational significance. We considered tunicamycin, thapsigargin, and Brefeldin A, as good tool compounds but ultimately excluded them from our study because of their minimal clinical potential.   

Bortezomib inhibits 20S proteasome catalytic activity and hence blocks the ER-associated protein degradation (ERAD) which in turn causes misfolded proteins to accumulate in the ER leading to the ER stress. At the same time, proteasome blockade affects degradation of hundreds of different proteins both in the cytoplasm and in the nucleus. Thus, I assume that a care should be taken when proteasome inhibitors are considered as specific ER-stress inducers;

Thank you for this point of clarification. A major mechanism of Bortezomib is UPR induction, but it indeed has pleiotropic effects. We added a statement indicating this point in the revised manuscript text on page 26 of the Discussion section.

I assume that exact IC50 values obtained following the incubation of compounds with CTCL cells should be indicated in the main text;

We have added IC50/EC50 values in the revised Figure 1C and revised text to include this change.

I would recommend putting the sequences of primers and antibodies into Tables;

Antibodies and primer sequences have been transferred to Tables as recommended.

Please describe the acquisition of HSPA6 FL, HSPA5 FL, and the HSPA5 substrate binding domain (SBD) in Materials and methods;

This has been added to the Materials and Methods section on pages 12-13.

Please describe fluorescence polarization (FP) assay in Materials and methods;

This has been added to the Materials and Methods section on page 13.

Was the sole effect of JG-023 on cell viability addressed?

Yes, for experiments examining synergy, the sole effect of JG-023 was addressed by normalizing DMSO (treatment control, no Btz) and JG-023 (no Btz) to 100%. This allows for true evaluation of the effect of JG-023 on the sensitivity of cells to Btz. Of note, the sole effect of JG-023 by itself was negligible at the concentration used (10 mM). A clarifying point about this methodology and references to previous publications where it was used has been added to the Materials and Methods section on page 12.

Perhaps, authors can benefit from using SynergyFinder 3.0 software (https://synergyfinder.fimm.fi);

Line 369 “with IC50s of 0.5±0.3, 1.3±0.2, and 1.4±0.06, respectively” please indicate the concentration;

A fixed 20 nM concentrations of FAM-JG-023 was used for the competition studies. This is provided on page 19 and in the Figure Legend for Fig. 3C.

Fig.1B several error bars are missing;

The error bars are present but not visible because the height of the symbol is larger than the value of the error.

Fig.5 A,B, please indicate statistically significant changes;

This has been added as requested in Figs. 5A-B and indicated in the figure legend. We used a one-way ANOVA to compare Normal CD4+ T cells to both HH and Hut78 CTCL cells.

Fig.5 C Please correct CD4+ to Normal CD4+;

Corrected. Please see updated manuscript Figure 5.

Line 443-444 “In normal CD4+ T cells, JG-023 significantly reduced Th2 type cytokines, including IL4, IL5, IL6, along with IL22” however, there is no visible difference in IL5 and IL22 gene expression in normal CD4+ cells (Fig. 5C). From the Figure 5C one can deduce that JG023 suppresses both Th1 and Th2 cytokine expression in CTCL cells;

The lack of a visible difference is due to the color gradient of the scale. In normal stimulated CD4+ T cells, there was a 9-fold induction of mRNA for both IL5 and IL22 which was reduced to zero with the addition of JG-023.

We address the suppression of both Th1 and Th2 cytokine expression in CTCL cells in the manuscript on page 19 lines 415-417. The conclusion we drew from this data was that broadly, for both normal and malignant T cells, inhibition of HSPA5/6 suppresses the Th2 cytokine signature which is known to play an important role in the progression of the disease and the response to therapy. 

Fig. 6 B, D, please indicate statistically significant changes.

Statistically significant changes added to Fig. 6B and accompanying legend.

Fig. 6D was modified as recommended below.

Please comment on zero levels of IFNg or IL-2 in Hut78 CM despite certain mRNA content shown in Figure 5A.

The analysis in Fig. 5A was conducted on PBMCs from normal donors, which were used as a source of normal T cells. In the unstimulated group (“-“) we consistently see negligible levels of IFNg until we stimulate them with the trimeric anti-CD3/28/2 antibody.

I suggest modification of Fig6. D;

Thank you for this suggestion. We have modified the figure by converting to heat map format, which more effectively captures the multiple variables in this set of experiments and further emphasizes the significance of IL-10 production in the immunosuppressive activity of CTCL cells. We have updated the figure, figure legend and manuscript text on page to reflect this change.

Line 476-478 “We found that CM from stimulated HH and Hut78 cells significantly repressed IFNg production following activation of normal T cells from three separate normal human donors in a dose dependent manner (Fig. 6A-B)” Were the PBMCs used in the experiment shown in Figure 6A also obtained from three different donors?

Figure 6A is representative data from a single donor experiment using a single dilution of HH and Hut78 CM. Figure 6B then evaluates suppression of T cell activation using PBMCs from multiple donors and different dilutions of CM.